

# Generalized analytical body force model for actuator disc computations of wind turbines

Jens N. Sørensen

Department of Wind and Energy Systems, Technical University of Denmark, Lyngby, 2800, Denmark

*Correspondence to*: Jens N. Sørensen (jnso@dtu.dk)

**Abstract.** A new generalized analytical model for representing body forces in numerical actuator disc models of wind turbines is proposed and compared to results from a Blade Element Momentum (BEM) model.  The model is an extension of a previously developed load model, which was based on the rotor disc being subject to a constant circulation, modified for tip

and root effects, corresponding to an optimum design case. By adding a parabolic circulation distribution, corresponding to a solid-body approach of the flow in the near-wake, it is possible to take into account losses associated with off-design cases, corresponding to pitch regulation at high wind speeds.  The advantage of the model is that it does not depend on any detailed knowledge concerning the actual wind turbine being analysed, but only requires information about the thrust coefficient and tip speed ratio. The model is validated for different wind turbines operating under a wide range of operating conditions. The

comparisons show generally an excellent agreement with the BEM model even at very small thrust coefficients and tip speed ratios.

## 1   Introduction

The actuator disc concept has for many years been employed as a means to include body forces into the Navier-Stokes equations for rotor computations of both single rotors (e.g. Sørensen and Myken (1992), Sørensen and Kock (1995), Ammara

et al. (2002), Mikkelsen (2003), Jimenez et al. (2007)) and multiple rotors operating in wind farms (e.g. Porté-Agel et al. (2011), Nilsson et al. (2015), Stevens et al. (2018)). The simplest way of implementing body forces in the Navier-Stokes equations is to let them be prescribed either as constant loadings (Sørensen et al., (1999)) or as prescribed radial distributions (Simisiroglou et al. (2016)). However, if more detailed information regarding load distributions is required, it is needed to know the actual rotor geometry, i.e. the twist- and chord distributions, as well as airfoil type at each cross section, including





the lift and drag characteristics of the airfoils (Sørensen and Kock (1995), Wu and Porté-Agel (2011)). Besides this, it also requires information regarding the operational envelope of the rotor, i.e. the collective pitch setting of the rotor blade and tip speed ratio as function of incoming wind speed. In many cases, however, this information is not known, either because geometry and airfoil data are confidential or simply because the developer has not yet decided size and type of the turbines in the initial development phase of a wind farm. There is therefore a need for a method that in a simple way may represent the

rotor loading by body forces without prior knowledge of the wind turbine.

A systematic study on different ways to include body forces were carried out by van der Laan et al. (2015), who showed that knowing the details of the actual loading results in a more reliable computations of the wake than simply assuming some more or less arbitrary shapes. In the study, a load model was proposed based on using the dimensionless load data from a known wind turbine to scale the loading for other turbines. Such a method actually indicates that in dimensionless form, the rotor

loading from one turbine is not very different from any other turbine. This assumption also forms the background for the analytical load model proposed by Sørensen et al. (2019). The model of Sørensen et al. (2019) is based on the assumption that, except near to root and the tip, the circulation is constant along the rotor blade. This makes it possible to derive an analytical set of equations describing axial and tangential load distributions along the blade that only depends on tip speed ratio and thrust coefficient. As a part of the model the load at the tip is modified by the usual tip correction (Glauert, 1935) and the root is

corrected by a polynomial. The model was validated using Large Eddy Simulation (LES) actuator disc computations of the Tjæreborg turbine and the DTU 10MW reference rotor operating at wind speeds corresponding to the design conditions. A further study employing the model on commercial wind turbine rotors operating at off-design conditions was recently performed by Sørensen and Andersen (2020). The studies showed that the analytical model performs excellently at design wind speeds (i.e. at wind speeds below the rated), whereas the load distributions start to deviate from the reference distributions

when operating the turbine away from the design load case. To complete the analytical load model to cope with the full range of wind speeds encountered by a wind turbine, there is therefore a need for a generalized extended version of the model. The aim of the present investigation is to devise a technique to extend the analytical load model to comprise wind turbines running under different operating regimes, including off-design conditions.

The paper is organized as follows. In section 2, the idea behind the analytical model is explained and the resulting set of

equations is derived. Section 3 contains a verification and tuning of the model parameters, and in section 4 results are presented and discussed. Section 5 contains a discussion of the results and the conclusion is given in section 6.

## 2   Methodology

In this section the derivation of the analytical body force model will be described in detail. First, the control strategy of a modern wind turbine is introduced in order to state the background for the extended version of the model, and next the equations

forming the analytical load model will be derived.



## 2.1 Power and thrust coefficients

To derive an analytical load model that is applicable for all wind speeds, it is required to understand the control strategy of a modern wind turbine. Most turbines of today are tip-speed regulated at wind speeds ranging from the cut-in wind speed to the

rated wind speed, which is the wind speed where the produced power becomes equal to the installed generator power. At higher wind speeds, the rotor is pitch-regulated in order to keep a constant power output. This involves turning the rotor blades about their long axis using an active control system that senses the blade position and at the same time measures the produced power to give the appropriate instructions for changing the blade pitch. The idea of pitch regulation is to limit the lift by decreasing locally the angles of attack on the blade.

As shown by Sørensen and Larsen (2021), the power production of a wind turbine at a given ambient mean wind speed, $U_0$, may below rated wind speed be approximated by the following generic expression

$$P(U_0) = \alpha U_0^3 + \beta \,, \tag{1}$$

where the coefficients $\alpha$ and $\beta$ are determined as

$$\alpha = \frac{P_G}{U_r^3 - U_{in}^3} \,, \qquad \beta = -\frac{P_G U_{in}^3}{U_r^3 - U_{in}^3} \,, \tag{2}$$

with $P_G$ denoting the rated (installed) generator power, $U_{in}$ is the cut-in wind speed and $U_r$ is the rated wind speed. This expression obviously allows for zero turbine production at the cut-in wind speed. The thrust and power coefficient are defined

as

$$C_T \equiv \frac{T}{\frac{1}{2}\rho A_R U_0^2} \,, \quad C_P \equiv \frac{P}{\frac{1}{2}\rho A_R U_0^3} \,, \tag{3}$$

where $T$ is the axial force, or thrust, acting on the rotor and $P$ is the power generated by the rotor, $\rho$ is the air density and $A_R = \frac{\pi}{4} D^2$ is the rotor area, with $D$ denoting the rotor diameter. We assume that the wind turbine operates at its optimum (rated) condition, $C_P = C_{P,r}$, at wind speeds lower than the rated wind speed, $U_r$, and at a constant power yield, $P = P_G$, at wind speeds higher than the rated wind speed. This operational strategy is typical for a modern wind turbine, which is



operated with a variable tip speed at wind speeds below the rated one, and which is pitch-regulated at higher wind speeds. An

example of this is illustrated in Fig. 1, which shows the performance curve of a 1500 kW UP1500-86 wind turbine from

Guodian United Power. It is here seen that the change from tip speed regulation to pitch regulation takes place at a wind speed

of about 10 m/s.

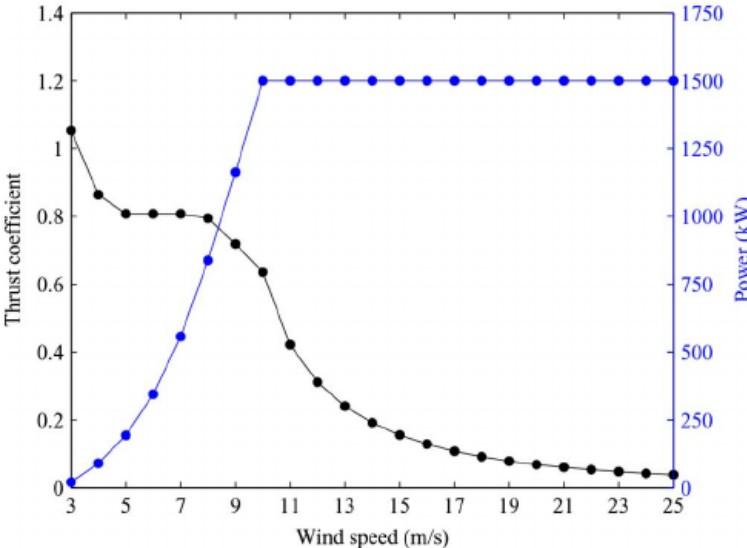

**Fig. 1: Typical power and thrust coefficient curves for a modern (1500 kW) wind turbine (from Gu et al., 2015).**


The rated wind speed is determined from eq. (3) at the condition where the generator operates at both maximum power and

maximum (rated) power coefficient,

$$U_r = \sqrt[3]{\frac{8P_G}{\rho\pi D^2 C_{P,r}}} \ . \tag{4}$$


With the above given assumptions, the wind turbine power curve is expressed as

$$P(U_0) = \begin{cases} \alpha U_0^3 + \beta & ; \ U_{in} \leq U_0 < U_r \\ P_G & ; \ U_r \leq U_0 \leq U_{out} \end{cases}, \tag{5}$$

where the wind turbine cut-out wind speed is denoted as $U_{out}$. The corresponding thrust coefficient, $C_T$, is approximated as





$$C_T = \begin{cases} C_{T,r} & ; \quad U_{in} \leq U_0 < U_r \\ C_{T,r} \cdot (U_0 / U_r)^{-3.2} & ; \quad U_r \leq U_0 \leq U_{out} \end{cases}.$$
(6)

From this expression it is seen that the thrust coefficient for $U_r \leq U_0 \leq U_{out}$ decreases with the wind speed to the power of -3.2. This value was recently derived analytically in a work by van der Laan et al. (2022). If not known in advance, typical values such as $C_{T,r} = 0.8$ and $C_{P,r} = 0.48$ may be employed to characterize the wind turbine performance.

**2.2 Basic equations of the load model**

Applying the Bernoulli equation in a rotating frame of reference across the rotor plane, we get the following expression for the pressure drop over the rotor disc,

$$\Delta p = \rho \Omega r u_\theta + \tfrac{1}{2} \rho u_\theta^2$$
(7)

where $\Omega$ is the angular velocity of the rotor, $u_\theta$ is the azimuthal velocity in the wake just behind the rotor, and r is the radial distance to the point considered. From this equation and the moment of momentum equation, sometimes referred to as Euler's turbine equation, we get the following two equations for the surface forces acting on the actuator disc representing the wind turbine:

$$f_z = \rho u_\theta (\Omega r + \tfrac{1}{2} u_\theta),$$
(8a)

$$f_\theta = \rho u_D u_\theta$$
(8b)

where $f_z$ and $f_\theta$ are the axial and azimuthal surface forces, respectively, and $u_D = u_D(r)$ denotes the axial velocity in the plane of the rotor. In dimensionless form, the equations read

$$\frac{f_z}{\tfrac{1}{2}\rho U_0^2} = \frac{u_\theta}{U_0}\left(2\lambda x + \frac{u_\theta}{U_0}\right)$$
(9a)

$$\frac{f_\theta}{\tfrac{1}{2}\rho U_0^2} = 2\frac{u_D}{U_0}\frac{u_\theta}{U_0}$$
(9b)

where $x = r/R$ is the dimensionless radial position $\lambda = \Omega R / U_0$ is the tip speed ratio.

The total axial force (thrust) and the power are determined by integration of the above equations,

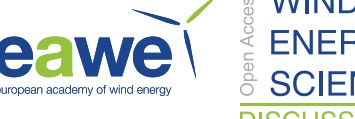

$$T = \int_0^R f_z \, 2\pi r dr = 2\pi\rho \int_0^R r u_\theta \left( \Omega r + \tfrac{1}{2} u_\theta \right) dr \tag{10a}$$

and

$$P = \Omega \int_0^R r f_\theta \, 2\pi r dr = 2\pi\rho\Omega u_D \int_0^R u_\theta r^2 dr , \tag{10b}$$


which in dimensionless form reads

$$C_T = \frac{T}{\tfrac{1}{2}\rho\pi R^2 U_0^2} = 4\lambda \int_0^1 \frac{u_\theta}{U_0} x^2 dx + 2\int_0^1 \left( \frac{u_\theta}{U_0} \right)^2 x dx \tag{11a}$$

$$C_P = \frac{P}{\tfrac{1}{2}\rho\pi R^2 U_0^3} = 4\lambda \int_0^1 \frac{u_D}{U_0} \frac{u_\theta}{U_0} x^2 dx \tag{11b}$$


From the above relations it is seen that a closure of the equations only demands knowledge about the azimuthal velocity distribution immediately behind the turbine.

For a wind turbine operating below rated wind speed, it is assumed that the rotor loading corresponds to the one obtained for an optimum rotor. Although the design of modern wind turbine rotors is based on different design objectives and

constraints, the actual geometry do generally not vary much. This assumption is supported from previous analyses by comparing optimum blade geometries generated using different rotor models. In Sørensen (2016) and Sørensen et al. (2021) a comparative study showed that for tip speed ratios typically used for the design of modern wind turbines, the different design methodologies approximately resulted in the same blade geometries. The idea behind the analytical model developed by Sørensen et al. (2019) is that an optimum designed blade is achieved by representing the rotor load by a constant circulation,

modified with a tip correction, $F(r)$, and a root correction, $g(r)$. However, when operating a turbine a wind speeds higher than the rated one, it becomes necessary to reduce the loading by regulating the pitch setting in order to maintain a constant power output. The impact of this is a redistribution of the loading, which no longer can be represented by a constant circulation. Since the difference in loading from the optimum one will create additional losses in the wake, the azimuthal velocity distribution forming the loading in eqs. (9a) and (9b) needs to include terms taking this into account. In the proposed model, the azimuthal

velocity distribution is not only represented by the induction from root and tip vortices, but also includes a term corresponding to a solid body rotation of the wake. When pitching the rotor this term will be active and ensure that the model includes the wake losses generated by the pitch setting. Hence, in the new extended model, the azimuthal velocity distribution is given as

$$\frac{u_\theta}{U_0} = \left[ \frac{q_0}{x} + S_0 x \right] g(x) F(x) , \tag{12}$$

where the first term in the bracket corresponds to the optimum constant circulation condition and the next term defines the redistribution of circulation due to the change in pitch setting when operating at wind speeds higher than the rated. The $q_0$





and $S_0$ are constants representing the circulation of the optimum rotor and the rotation of the solid body term, respectively. As tip correction we employ the model proposed by Glauert (1935),

$$F = \frac{2}{\pi} \arccos\left[ \exp\left( -\frac{N_b(1-x)}{2\sin\phi} \right) \right], \tag{13}$$

where $N_b$ denotes the number of rotor blades and $\phi$ is the local flow angle, which approximately can be determined from the formula,

$$\sin\phi = \frac{1}{\sqrt{1 + \lambda^2 x^2 / \left(\dfrac{u_D}{U_0}\right)}}. \tag{14}$$

It should be mentioned that we in the original work (Sørensen et al., 2019) used the tip correction of Prandtl. However, to be consistent with usual standards in BEM theory, this is replaced by the tip correction of Glauert. To account for the influence of the hub and the inner non-lifting part of the rotor, a vortex core of size $\delta$ is introduced, and an expression for the root correction is proposed as follows,

$$g = 1 - \exp\left[ -a\left( \frac{x}{\bar{\delta}} \right)^b \right], \tag{15}$$

where $\bar{\delta} = \dfrac{\delta}{R}$ denotes the dimensionless radial distance to the point where the maximum azimuthal velocity is achieved. With the proposed model, $\bar{\delta}$ typically corresponds to the point where the lifting surface of the rotor starts. In the general case, the relation between the constants $a$ and $b$ is determined by differentiating eq. (15) to determine the maximum azimuthal velocity at $x = \bar{\delta}$. Here we assume $g$ to be represented by a 4th order polynomial, hence we get the values $b=4$ and $a=2.335$. A derivation of the general relationship between $a$ and $b$ is given in Appendix A.

Inserting eq. (12) into eqs. (9a) and (9b), the following expressions are obtained

$$\frac{f_z}{\frac{1}{2}\rho U_0^2} = \left( 2\lambda x + \left( \frac{q_0}{x} + S_0 x \right) g(x) F(x) \right) \left[ \frac{q_0}{x} + S_0 x \right] g(x) F(x) \tag{16a}$$

$$\frac{f_\theta}{\frac{1}{2}\rho U_0^2} = 2\frac{u_D}{U_0} \left[ \frac{q_0}{x} + S_0 x \right] g(x) F(x) \tag{16b}$$

Inserting eq. (16a) into eq. (11a), we get





$$C_T = \int_0^1 2x \left( 2\lambda x + \left( \frac{q_0}{x} + S_0 x \right) g(x) F(x) \right) \left[ \frac{q_0}{x} + S_0 x \right] g(x) F(x) dx \implies$$

$$C_T = 2a_1 q_0^2 + 4a_2 \lambda q_0 + 4a_3 S_0 q_0 + 4a_4 \lambda S_0 + 2a_5 S_0^2$$

$(17)$


The coefficients from the integration are given as follows,

$$a_1 = \int_0^1 \frac{g^2 F^2}{x} dx \, ; \quad a_2 = \int_0^1 gFx dx \, ; \quad a_3 = \int_0^1 g^2 F^2 x dx \, ; \quad a_4 = \int_0^1 gFx^3 dx \, ; \quad a_5 = \int_0^1 g^2 F^2 x^3 dx \, .$$

From eq. (17), the dimensionless reference circulation is determined as

$$q_0 = \frac{\sqrt{\left( a_2 \lambda + a_3 S_0 \right)^2 + a_1 \left( \tfrac{1}{2} C_T - 2a_4 \lambda S_0 - a_5 S_0^2 \right)} - \left( a_2 \lambda + a_3 S_0 \right)}{a_1} \, .$$

$(18)$

Inserting eq. (16b) into eq. (11b), we get


$$C_P = 4\lambda \int_0^1 x^2 \frac{u_D}{U_0} \left[ \frac{q_0}{x} + S_0 x \right] g(x) F(x) dx \implies$$

$$C_P = 4\lambda \left( a_6 q_0 + a_7 S_0 \right)$$

$(19)$

where $a_6 = \int_0^1 \frac{u_D}{U_0} gFx dx$ and $a_7 = \int_0^1 \frac{u_D}{U_0} gFx^3 dx$ .


It is seen that eq. (19) allows for solutions involving non-constant inflow like shear or yaw. In Sørensen et al. (2019) it was demonstrated how arbitrary inflow velocity distributions can be included to determine local load distributions. However, in the present work, where the focus is on validating the basic approach, we assume a constant inflow. In this case, we get



$$C_P = 4\lambda \frac{u_D}{U_0}\left(a_2 q_0 + a_4 S_0\right) . \tag{20}$$

Knowing the power coefficient, the axial velocity in the rotor plane is given as

$$\left(\frac{u_D}{U_0}\right) = \frac{C_P}{4\lambda\left(a_2 q_0 + a_4 S_0\right)}. \tag{21}$$

The above described system of equations forms the basis of the proposed analytical load model. Input to the model is $(\lambda, C_T, C_P)$ or, alternatively, $(\lambda, C_T, u_D/U_0)$, depending on the aim of the analysis.

At wind speeds below the rated, the rotor is operated at a constant tip speed ratio, hence $S_0 = 0$ and the reference circulation is only a function of rated tip speed ratio, thrust coefficient and power coefficient, $q_{0,r} = q_0(\lambda_r, C_{T,r}, C_{P,r})$. From eq. (18), we get

$$q_{0,r} = \frac{\sqrt{\lambda_r^2 a_2^2 + \tfrac{1}{2} a_1 C_{T,r}} - \lambda_r a_2}{a_1} . \tag{22}$$

With the axial flow in the rotor plane computed from

$$\left(\frac{u_D}{U_0}\right)_r = \frac{C_{P,r}}{4\lambda_r a_2 q_{0,r}} . \tag{23}$$

Hence, setting $S_0 = 0$, the dimensionless loading can be obtained directly as in the original analytical model derived by Sørensen et al. (2019). In Sørensen and Andersen (2020) this approach was shown to give excellent results for rotors operating at rated conditions. However, at operating conditions far from the rated, the assumption of a constant circulation supplemented with tip and root corrections was found not to be sufficient, and an extended modelling, as the one proposed here (eq. 12), is required. In this context, two main questions remain to be answered. First, does the proposed extended model, eqs. (16a) and (16b), actually represent the loads on a real rotor operating at off-design conditions? Secondly, since an additional parameter, $S_0$, is introduced, an additional relationship connecting this parameter and the existing input parameters is required in order to establish a solution at off-design. How do we model this? These two questions will be addressed in the following section.



## 3 Verification and tuning of model

In this section, we verify the basic behaviour of the proposed load model and tune the modelling parameter $S_0$.

### 3.1 Verification of the model at off-design conditions

A simple way to verify the applicability of the proposed model to represent the loadings at off-design conditions is to compare it to results obtained from Blade-Element Momentum (BEM) computations of an actual wind turbine. Since the parameter, $S_0$, is not known, we simply try different values and chose the one that gives the best fit of the analytical load distributions to those computed by the BEM technique. As a reference, we chose the geometry of the NEG Micon NM80 wind turbine, which in the previous study by Sørensen and Andersen (2021) was employed to test the model for $S_0 = 0$ (more details about the

NM80 turbine is given in App. B). We here chose three different operating conditions, one corresponding to the rated condition ($C_T = 0.82$) and two off-design conditions ($C_T = 0.26$ and $C_T = 0.13$).

In the following we compare dimensionless normal and tangential load distributions along a blade. The distributions are normalized by $\rho R U_0^2$, hence the dimensionless quantities are given as,

$$C_n = \frac{F_n \, [N/m]}{\rho R U_0^2}; \qquad C_t = \frac{F_t \, [N/m]}{\rho R U_0^2} \, , \qquad (24)$$

where $F_n$ is the normal loading and $F_t$ is the tangential loading on each blade. As the analytical loads in eqs. (9a) and (9b) are given per area unit for the full rotor, the load coefficients are computed as

$$C_n = \left( \frac{f_z}{\frac{1}{2}\rho U_0^2} \right) \cdot \left( \frac{\pi x}{N_b} \right); \qquad C_t = \left( \frac{f_\theta}{\frac{1}{2}\rho U_0^2} \right) \cdot \left( \frac{\pi x}{N_b} \right) \, , \qquad (25)$$

where $N_b$ denotes the number of blades.

The results are shown in Fig. 2, which depicts distributions of normal loadings (left) and tangential loadings (right). In the plots, the BEM computations are given as solid lines and the analytical results are given symbols (a triangle for $S_0 = 0$ and

a circle for the 'optimum' $S_0$-value). First, it is observed that the comparison between BEM and the analytical model at rated



conditions (black curves) displays an excellent agreement. Here the optimum $S_0$-value is zero, confirming that the original model actually is working well for the case at which it is developed. In the two off-design cases, on the other hand, it is clearly seen that it is required to extend the model with an additional term. In particular, the tangential loading becomes way off if we maintain a value $S_0 = 0$, where the distributions tends to keep a nearly constant tangential load distribution over the main

part of the rotor blade. Since we have no expression to determine $S_0$, we try different values and choose the one that gives the distributions most close to the BEM computations. In the present case these are found to be $S_0 = 0.019$ for $C_T = 0.26$ and $S_0 = 0.044$ for $C_T = 0.13$. Employing these values, the comparison displays an excellent agreement for both the normal and tangential load distributions, demonstrating that the proposed model actually takes into account the main features of the load distributions at off-design conditions. However, it is still needed to develop a general expression for the rotation parameter $S_0$

265 .

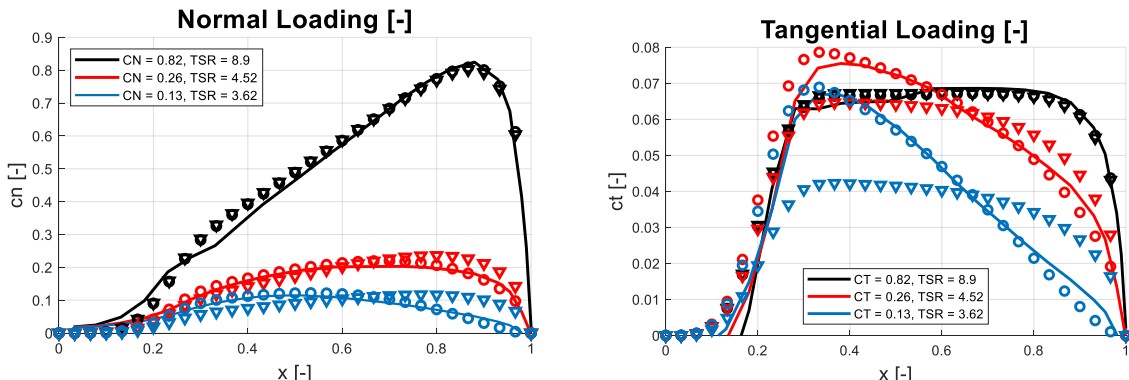

**Fig. 2: Comparison between analytical model and BEM computations. Solid lines: BEM computations; Triangles:**
**Analytical model assuming $S_0 = 0$; Circles: Analytical model using 'Optimized' $S_0$-values. Left figure: Normal load distributions; Right figure: Tangential load distributions.**

**3.2 Modelling of the rotation parameter $S_0$**

To determine an expression for the rotation parameter $S_0$, we first recognize that $S_0$ is equal to zero for a rotor operating at rated conditions. Hence, it is natural to seek for an expression that depends on how far the rotor is operating from the rated one, i.e. search for a relationship $S_0 = S_0(\lambda - \lambda_r)$ or $S_0 = S_0(C_T - C_{T,r})$, where $\lambda_r$ and $C_{T,r}$ denote the rated values of

the tip speed ratio and the thrust coefficient, respectively. To accomplish this, as a starting point we carry out a series of BEM computations for actual wind turbines operating at different conditions. The used wind turbines are the Vestas V27 and V52 turbines and the NEG Micon NM80 turbine. The relevant data for the turbines are given in Appendix B. The computations are



carried out at different operating conditions and the outcome is employed to determine if there exists a simple parameterizable

relationship between $S_0$ and the input variables. Hence, for each combination of $\lambda$ and $C_T$, a BEM computation of the

normal and tangential loading distributions is carried out and the value of $S_0$ that best fits the distributions with the analytical

model is determined. The outcome of this is for the tree wind turbines and different combinations of tip speed ratio and thrust

coefficients shown in Fig. 3, where $S_0$ is plotted as a function of normalized values of the tip speed ratio, $(\lambda_r - \lambda)/\lambda_r$, and

thrust coefficient, $(C_{T,r} - C_T)/C_{T,r}$. Analyzing the two distributions, it is seen that using the normalized tip speed ratio to

determine $S_0$ results in some scatter of the data, whereas the there is a more

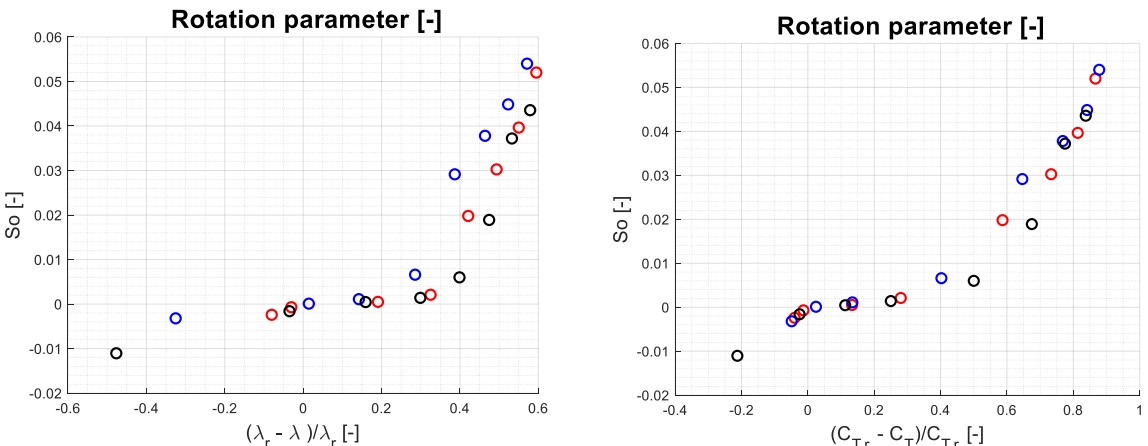

**Fig. 3: Correlation between rotation parameter, $S_0$, and dimensionless relative tip speed ratio (left) and thrust coefficient (right). Red circles: V27; Blue circles: V52; Black circles: NM80.**

unique relationship between   and the normalized thrust coefficient.  Employing a simple least squares fit using the relationship

$$S_0 = \alpha \left( \frac{C_{T,r} - C_T}{C_{T,r}} \right)^{\beta}, \qquad (26)$$

we get the values $\alpha = 0.08$ and $\beta = 3$ for $C_T < C_{T,r}$ and $\alpha = 0.05$ and $\beta = 1$ for $C_T \geq C_{T,r}$. The result of the fit is

shown in Fig. 4, which shows a very good agreement between the computed points

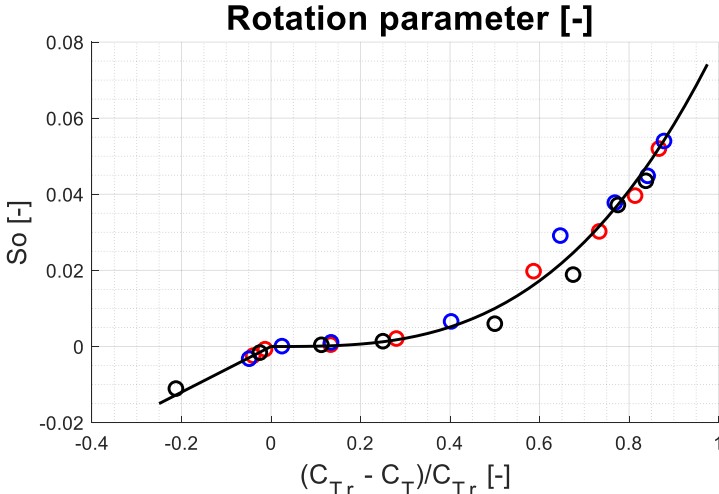

**Fig. 4: Curve fit given the parameterization of $S_0$ with respect to the normalized thrust coefficient. Red circles: V27; Blue circles: V52; Black circles: NM80.**

and the curve fit. Hence, the closure of the equations is accomplished by exploiting the below expression to connect the rotation parameter $S_0$ with the normalized thrust coefficient:

$$
S_0 = \begin{cases} 0.08 \cdot \left( \dfrac{C_{T,r} - C_T}{C_{T,r}} \right)^3 & ; \quad C_T < C_{T,r} \\[2em] 0.05 \cdot \left( \dfrac{C_{T,r} - C_T}{C_{T,r}} \right) & ; \quad C_T \geq C_{T,r} \end{cases}
. \tag{27}
$$

As a conclusion of the parameterization, we now have a closure of $S_0$ at off-design conditions that besides the actual thrust

coefficient, $C_T$, also demands knowledge about the rated thrust coefficient, $C_{T,r}$. For an actual wind turbine, this value is normally given as a part of the general technical data. If this is not known, a value of 0.8, which is typical for a commercial wind turbine, may be employed.

## 4    Results

In the following we show various comparative results for three different wind turbines operating at a broad range of conditions.

The BEM computations are carried out using full knowledge regarding the actual blade geometry and associated airfoil data. The operational conditions contain all kinds of combinations between the tip speed ratio and the thrust coefficient, including





high wind speeds, corresponding to low thrust coefficients. In this case, the thrust coefficient is lowered by pitching the rotor blades. In contrast to the detailed input required in the BEM computations, the analytic model only demands tip speed ratio, thrust coefficient and power coefficient as input. The computations are carried out for a constant axial inflow without shear

and turbulence. As demonstrated in Sørensen et al. (2019), shear and turbulence are easily introduced into the model when carrying out actual CFD/actuator disc computations, but it is not the objective of the present work to include this. Here we focus on assessing the models ability to represent loadings for different turbines operating at off-design conditions. The chosen wind turbines represent sizes with a range of nameplate capacity from 225 kW to 2750 kW. The data for the wind turbines are given below in Appendix B, which shows nameplate capacity, rotor diameter and design tip speed ratio. The latter information

is included to assess where the best agreement between the analytical model and the BEM computations can be expected to take place.

In Figs. (5) – (7) the load distributions are compared for three wind turbines operating at a wide range of off-design conditions with thrust coefficients ranging from 0.1 to 0.97 and tip speed ratios between 3 and 13, where the turbine is either pitch-regulated or running at upstart conditions. The only input to the analytical model are the tip speed ratio and the $C_T$ and

$C_P$ values from the BEM computations. Fig. (5) shows comparative normal and tangential force distributions for the V27 rotor. As seen, there is a generally a very good agreement between the analytical and the BEM computed normal force distributions. The agreement between analytical and computed tangential force coefficients is good over most of the rotor surface, but not as convincing as for the normal force distributions. The biggest deviation between computed and analytical tangential loadings is seen to appear at the inner part of blade, and, for some unknown reason, the best comparisons are for

high and low $C_T$-values. Fig. 6 compares computed and analytical force coefficients for the V52 wind turbine. As compared to the V27 data, we here observe an even better agreement between analytical and computed force distributions. In particular, the comparison between the analytical representation of the tangential force distribution and the computed one is excellent for all the predicted cases. However, some deviation is observed for the normal force coefficients at the outer part of the blade. The best comparison between the analytical and the computed force coefficients are found for the NM80 turbine, as shown in

Fig. 7. We here observe an excellent agreement between the analytical and numerical curves for all $C_T$-values.





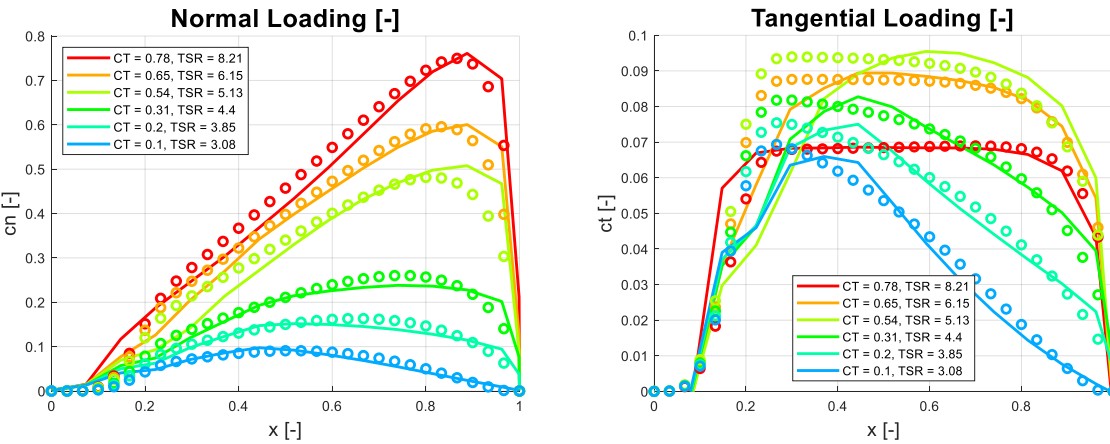

**Fig. 5: Normal and tangential force coefficient distribution of the V27 turbine at different tip speed ratio and thrust coefficient. Circles: Analytical model; Solid lines: BEM computations.**


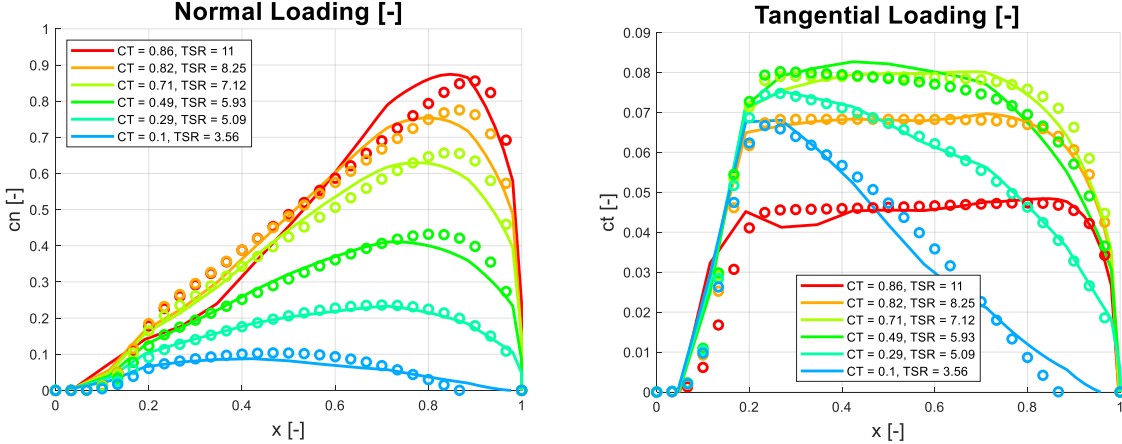

**Fig. 6: Normal and tangential force coefficient distribution of the V52 turbine at different tip speed ratio and thrust coefficient. Circles: Analytical model; Solid lines: BEM computations.**



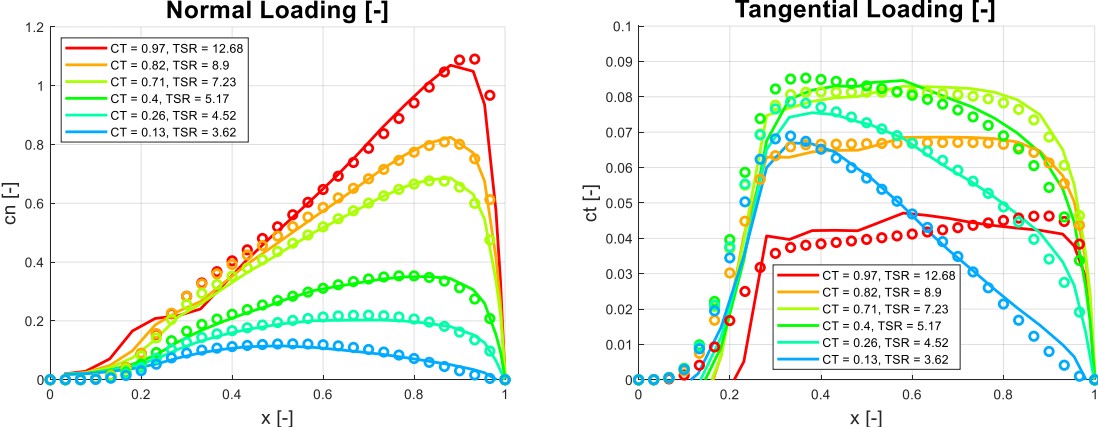

**Fig. 7: Normal and tangential force coefficient distribution of the NM80 turbine at different tip speed ratio and thrust coefficient. Circles: Analytical model; Solid lines: BEM computations.**

## 5 Discussion

The comparison between the BEM computations and the analytical load distributions are generally in very good agreement with each other, even for wind turbines operating far away from the rated design-based conditions. This actually support the underlying hypothesis that there exists a general way of describing the loading on a wind turbine using a simple analytical expression with only very few input parameters. For the present investigation, input parameters are the thrust and power coefficients as function of tip speed ratio. In a large eddy simulation of e.g. a wind farm, the velocity distribution on the actuator disc (i.e. the rotor) is an inherent part of the simulation and the unknown in this case is the generated power (see e.g Nilsson et al. (2014) or van der Laan et al. (2015)). There may be other ways of representing and generalize the expressions for the loadings. The one proposed here, using circulation and rotation of the wake flow as a guideline to parameterize the loads, seems actually to work quite well. One may argue that using the BEM technique, which normally is characterized as a low-fidelity approach, as basis for determining the missing relationship between the $S_0$-parameter and the thrust coefficient is not accurately enough. An answer to this is partly that the BEM technique, even today, is the only design tool used in industry for designing wind turbine rotors, and partly that the methodology presented here is general and the parameterization and tuning of the model easily can be improved later using more sophisticated prediction tools.

## 6 Conclusion

A generalized analytical body force model has been developed and validated against load distributions generated by a BEM model. The model, which is an extension of an earlier model that was only valid for optimum operating rotors, now includes load expressions for wind turbine rotors operating at off-design conditions. The essential part of the model is based on combining an expression for constant circulation with a solid body rotation approach to take into account losses when operating





the rotor at off-design conditions. A comparison with BEM computations was carried out using three wind turbines of different sizes running at a range of different operating conditions. The results are very convincing, showing generally a very good agreement between the simple analytical model and the BEM results. The comparison demonstrate that a simple analytical model with very good precision can be utilized to represent the loading on wind turbines, both at design and off-design

conditions.

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

**Appendix A**

The expression for the root correction is derived from the idea that the inner part of the rotor is a viscous correction to the potential vortex forming the lift-producing part of loading. At design conditions (where $S_0 = 0$), the velocity azimuthal 425 distribution near the root is given as

$$\frac{u_\theta}{U_0} = \frac{q_0}{x}\left[1 - \exp\left[-a\left(\frac{x}{\bar{\delta}}\right)^b\right]\right],$$
(A.1)

where we implicitly assume that $F = 1$. To determine the relationship between between the constants $a$ and $b$ we assume that the azimuthal velocity attains its maximum at the radial position where $x = \bar{\delta}$. Hence, the relation between $a$ and $b$ is 430 determined by differentiating eq. (A.1) with respect to x and setting this expression equal to zero at $x = \bar{\delta}$. Differentiating eq. (A.1) with respect to $x$ gives

$$\frac{d}{dx}\left(\frac{u_\theta}{U_0}\right) = -\frac{q_0}{x^2}\left[1 - \exp\left[-a\left(\frac{x}{\bar{\delta}}\right)^b\right]\right] + \frac{q_0}{x}\exp\left[-a\left(\frac{x}{\bar{\delta}}\right)^b\right]\frac{ab}{\bar{\delta}}\left(\frac{x}{\bar{\delta}}\right)^{b-1}.$$
(A.2)





Inserting $x = \bar{\delta}$ and setting the expression equal to zero, gives the following relation between $a$ and $b$,

$$(a \cdot b + 1)\exp(-a) = 1. \tag{A.3}$$

We here observe that the expression does not include the viscous core size, $\bar{\delta}$, but is a generic expression for the relationship between the parameters $a$ and $b$. Since the equation is non-linear, it is required to solve it numerically. Doing this, the relationship between a and b is as shown in Fig. A1. In the present work, we put $b = 4$ and get that $a = 2.335$.

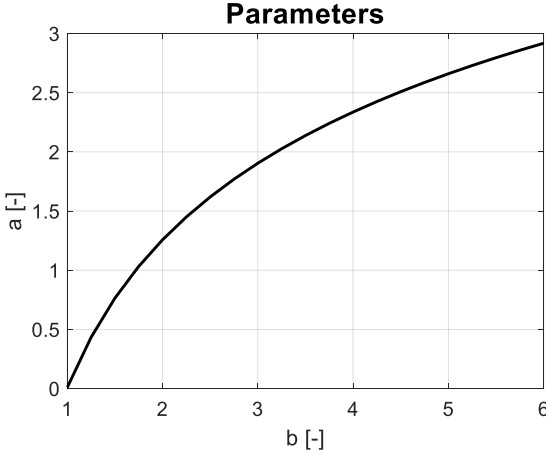

**Fig. A.1 Plot showing the relationship between the parameters $a$ and $b$ (eq. A.3).**

**Appendix B**

Here we give the main characteristics of the turbines used in the study. We do not present detailed data, such as chord- and twist-distributions, or the employed airfoil characteristics, as they are confidential. However, for the present study this data is not needed, since only the outcome of the BEM computations is required to develop and validate the developed model. Details of the Vestas V27 turbine can be found in Resor and LeBlanc (2014) and Kelley and White (2018), and for the V52 turbine, the reader is referred to the homepage https://en.wind-turbine-models.com/turbines/71-vestas-v52.

**Table 1. Wind turbine characteristics**

|  | Vestas V27 | Vestas V52 | NEG Micon NM80 |
|---|---|---|---|
| Name plate capacity [kW] | 225 | 850 | 2750 |
| Diameter [m] | 27 | 52 | 80 |
| Design tip speed ratio [-] | 7.6 | 8.3 | 8.6 |