# Peer review of "Generalized analytical body force model for actuator disc computations of wind turbines"

_Wind Energy Science, 2022_

## Referee Comment (RC2)

[referee-annotated manuscript omitted]

---

## Author Comment (AC1)

**Author reply to reviewer comments**

**Reviewer 1**

Leonardo DaVinci once said: "Simplicity is the ultimate sophistication." This seems to readily apply to the proposed generalized analytical body force model presented in the manuscript. Great work that has potential for significant impact. Well done.

Scientific Significance: The manuscript presents a generalized actuator disc (AD) model for use in wind farm simulations. The novel scientific aspect of the extended (i.e. generalized) model is that it enables AD methods for use at off-design (Region III) wind speeds, as opposed to be only valid for design (i.e. close to optimal) conditions in Region II. This merits not only publication in WES but also is potentially impactful to the broader community on wind farm wake modeling.

Scientific Quality: The mathematical methods used in the manuscript appear to be correct. All relevant information is given for the reader to implement the proposed model in independent analyses. The work is of the highest scientific quality, demonstrating physical insight into wind turbine aerodynamics at off-design conditions.

Presentation Quality: Good writing style w/o being excessive in the description of background, motivation, mathematical formulation, graphs etc. Very well done.

**Thank you very much for your very positive comments. I am happy to hear that the paper is well received.**

Minor Comments:

• Typos in lines 144, 282, 285, 326. Please doublecheck.

**Thanks for pointing at these typos. They have now been corrected accordingly.**

• Section 3.1: The author may consider revising the paragraph describing 'tuning' of the model. The reviewer would hope for a more physics-based explanation/description; at a minimum, what range of values are expected for S0 and why?

**I agree and an appendix (App. C) has been added that from physical arguments give the range of S0.**

• Page 14: Discussion of tangential loading seen in Figs. 5-7. The explanation in line 329 "for some unknown reason" is incomplete. Here again, the reviewer would expect some physics-based explanation for the observation. For example, is there a distinct difference in the design of the root region for the V27 in comparison to the V52 and NM80 ? If yes, then the behavior of the tangential loads can probably be explained. Please investigate a little further.

In the modified version of the paper the sentence 'for some unknown reasons' has been deleted. The differences seem not to be related to the geometry of the rotors, as the analytical results for all rotors fit very well with the BEM-computed loading at rated wind speed. Instead, the rotors may have different control strategies when going from region II to region III, which may explain the

differences. However, I am not aware of the details of this and therefore this will not be discussed in the paper.

**Reviewer 2**

This manuscript presents an extension of an existing analytical body force model for a Joukowsky rotor with constant bound circulation, previously developed by the author, to a contemporary pitch-regulated rotor with variable circulation by introducing a parabolic circulation distribution. The parameters for the parabolic circulation distribution were derived through the characterization of various wind turbine models. The accuracy of the final model was verified against Blade Element Momentum (BEM) results. This simple analytical model has potential utility for the wind energy industry.

I have two general comments regarding the manuscript:

1. To enhance readers' comprehension of the derivation in section 2.2, it may be beneficial to include a schematic plot with coordinates.

**Agreed. A schematic plot is now included in the modified version of the paper.**

2. As the parameter S0 is tuned on the three rotors of V27, V52, NM80. It would be great to present the verification results for a different wind turbine model to show the robustness of the analytical model.

Since the present work includes three different wind turbines of very different sizes, operating at all kinds of off-design conditions, I am actually confident of the robustness of the model. I agree that the model needs to be challenged by computations of the next generation of large wind turbines. This will be the subject of future work where we expect to exploit the model for LES computations of wind farms and as a part of this validate in more detail the model.

Please refer to the attached pdf file for some additional comments.

*Line 83: The equation defining* C*P*,*r* *is now included.*

Line 104: It is now stated that the suggested CT and CP values are for an 'actual' rotor (not an ideal rotor).

Line 110: Reference included for the derivation of eq. (7).

Line 155: More explanation regarding eq. (12) included.

Line 200: Text modified in order to explain why eq. (19) allows for non-constant inflow.

Good luck with the revision!

Thanks, and thanks for the good comments.